# Knowledge Gaps and Research Priorities on the Health Effects of Heatwaves: A Systematic Review of Reviews

**DOI:** 10.3390/ijerph19105887

**Published:** 2022-05-12

**Authors:** Andrea Conti, Martina Valente, Matteo Paganini, Marco Farsoni, Luca Ragazzoni, Francesco Barone-Adesi

**Affiliations:** 1CRIMEDIM—Center for Research and Training in Disaster Medicine, Humanitarian Aid and Global Health, Università del Piemonte Orientale, 28100 Novara, Italy; martina.valente@uniupo.it (M.V.); matteo.paganini@uniupo.it (M.P.); francesco.baroneadesi@uniupo.it (F.B.-A.); 2Department of Translational Medicine, Università del Piemonte Orientale, 28100 Novara, Italy; 20002321@studenti.uniupo.it; 3Department of Sustainable Development and Ecological Transition, Università del Piemonte Orientale, 13100 Vercelli, Italy

**Keywords:** climate change, global warming, extreme weather event, heatwave, public health

## Abstract

Although extreme weather events have played a constant role in human history, heatwaves (HWs) have become more frequent and intense in the past decades, causing concern especially in light of the increasing evidence on climate change. Despite the increasing number of reviews suggesting a relationship between heat and health, these reviews focus primarily on mortality, neglecting other important aspects. This systematic review of reviews gathered the available evidence from research syntheses conducted on HWs and health. Following the PRISMA guidelines, 2232 records were retrieved, and 283 reviews were ultimately included. Information was extracted from the papers and categorized by topics. Quantitative data were extracted from meta-analyses and, when not available, evidence was collected from systematic reviews. Overall, 187 reviews were non-systematic, while 96 were systematic, of which 27 performed a meta-analysis. The majority evaluated mortality, morbidity, or vulnerability, while the other topics were scarcely addressed. The following main knowledge gaps were identified: lack of a universally accepted definition of HW; scarce evidence on the HW-mental health relationship; no meta-analyses assessing the risk perception of HWs; scarcity of studies evaluating the efficacy of adaptation strategies and interventions. Future efforts should meet these priorities to provide high-quality evidence to stakeholders.

## 1. Introduction

Extreme weather events have plagued humanity since its inception [1]. Temperature extremes encompass heatwaves (HWs) and cold spells. In particular, HWs, namely periods of abnormally high temperatures, are progressively increasing due to climate change [2,3]. Mounting evidence shows that HWs have become more frequent and intense in the past seventy years and that they will become a common phenomenon in different areas of the world by the end of the century [4]. Ma et al. found a sevenfold increase in the likelihood of extreme heat over 1950–2014 due to anthropogenic climate change [5]. An increase in HW frequency, duration, and intensity was identified by Li in Southeast Asia among the last four decades [6]. Guerriero et al. projected an increase in HW days especially in cities in southern Europe, while cities in central Europe are expected to experience the greatest HW temperature increases [7].

This trend, in addition to the world population aging and the progressive urbanization, is expected to substantially increase the number of individuals vulnerable to the effects of high temperatures in the near future [8,9]. Indeed, between 2000 and 2016, the number of vulnerable individuals exposed to HWs increased by about 125 million [10], a trend fostered by the growth of the global population especially in hotter regions of the world [11]. This is particularly worrying, as HWs are among the extreme weather events that cause the largest losses in terms of human lives [12].

Scientific literature on the health effects of high temperatures has sharply increased in recent years. Although this trend regarded not only original studies but also secondary studies (i.e., reviews), it appears that not all aspects have been equally covered [13]. For example, some authors pointed out the relative scarcity of morbidity studies compared to the mortality ones and suggest that research in this field tends to revolve around selected aspects while overlooking other important areas that instead could allow a substantial advancement of knowledge [14].

The COP26 held in November 2021 and the last Assessment Report of the Intergovernmental Panel on Climate Change have underlined the need to adapt to the impacts of climate change as a crucial action point [15,16]. For this reason, it is necessary to thoroughly evaluate the current scientific evidence on the health effects of high temperatures, in order to identify knowledge gaps to be filled in the years to come. We carried out a systematic review of published reviews to systematically map all the available evidence on the short-term, acute impact of HWs on human health, thus addressing future research and policy directions and advancing research on climate change adaptation and resilience.

## 2. Methods

### 2.1. Study Design

Since the scientific literature has extensively investigated the health effects of HWs and there is already a heterogeneous body of reviews on this topic, a systematic review of published reviews was deemed the most appropriate methodology for mapping existing knowledge and identifying future research directions [17,18]. Preferred Reporting Items for Systematic Reviews and Meta-Analyses (PRISMA) was followed as a guide for conducting and reporting the present review [19].

### 2.2. Search Strategy

A systematic literature search was carried out on PubMed and Scopus databases. The search strings (Appendix A) combined two different blocks of terms, namely heat-related and health-related ones. The search, performed on 14 September 2021, was limited to articles published from the year 2000 onwards and was restricted to literature reviews. The study selection process relied on the following inclusion criteria: (a) the study was a review (systematic review with meta-analysis, systematic review without meta-analysis, non-systematic review); (b) the study was available in English, Italian, German, French, Portuguese, or Spanish; (c) the study evaluated the short-term, acute impact of high temperatures on human health. Exclusion criteria were: (a) the study dealt with indirect, long-term impacts of global warming (e.g., changing patterns of vector-borne diseases, food insecurity); (b) the study considered or described the pathophysiology and clinical features of the direct effects of sun exposure, heatstroke, or heat strain; (c) the study focused on heat effects on animals’ health or on the environment.

#### Data Extraction, Analysis, and Synthesis

Extracted details of the articles were inserted in a Google Sheet (Google, 1600 Amphitheatre Parkway, Mountain View, CA, USA) database and duplicates were removed. Three authors (AC, MP, MF) independently screened the articles, and inconsistencies were resolved after confronting with the whole group. The screening process aimed at assessing studies’ eligibility for inclusion, as well as classifying articles according to the type of review and the topic addressed. An operational categorization of the reviews was performed according to the following criteria: (a) the study was a systematic review with meta-analysis (hereinafter referred to as meta-analysis); (b) the study was a systematic review without meta-analysis (i.e., search strategy and screening process are performed through a systematic methodology explicitly declared in the text, but no quantitative summary of results was provided; hereinafter referred to as systematic review); (c) the study was a narrative review or a review performed without specifying any search strategy and screening criteria (hereinafter referred to as non-systematic review).

Data were extracted from the reviews following a combination of inductive and deductive approaches. The subjects covered by each review were systematically mapped into different *topics*. These topics were inductively gathered by the literature and were then grouped into broader *categories*. Specifically, three researchers (AC, MP, MV) independently extracted information from the reviews and categorized them according to an evolving list of topics. This identification process was recursively repeated and emerging topics were added to the list until saturation. The categorization was then discussed and consolidated upon reaching agreement within the research team. In detail, a review was considered as dealing with a topic whenever all the following conditions applied: (a) the topic was addressed in the text; (b) the topic was expressed in relation to HWs; (c) references to original studies dealing with the topic were reported in the review. Some reviews contributed to more than one topic. The topics were subsequently grouped into five different categories. Upon completion of data extraction, a total of five categories and 24 topics were identified (Figure 1).

## 3. Results

The search returned a total of 2232 records. After removing duplicates, 1176 titles and abstracts were screened, and 165 articles were excluded because they were not reviews. For 26 articles it was not possible to retrieve the full text. After reading the full text of the remaining ones, 1280 were excluded because the topic did not match the eligibility criteria and 22 further articles were excluded because written in a language different from those listed in the inclusion criteria. In total, 283 reviews were included in our analysis, as outlined in Figure 2.

The list of all the included reviews, as well as the classification within categories and topics, are shown in Appendix A. In total, 27 were meta-analyses, 69 were systematic, and 187 were non-systematic reviews. A temporal increase in all publication types can be noted in Figure 3.

The number of reviews dealing with the different topics differed largely, with mortality (number of records = 162), vulnerability (159) and morbidity (147) that were much more covered than adaptation (91) and risk perception (17). Figure 4 shows how this difference further increased over the years.

For every addressed topic, non-systematic reviews were the most common type of retrieved review. Meta-analyses were in general scant and were absent when it comes to the topics of risk perception and adaptation Appendix A.

In the following sections, evidence regarding the different topics is reported. Whenever possible, this was based on results of meta-analyses. When these were absent for a specific topic, evidence from systematic reviews is reported. Estimates that referred to different definitions of HW have been reported as well and distinguished in the text.

### 3.1. Mortality

#### 3.1.1. All-Cause Mortality

In total, four meta-analyses dealt with all-cause mortality [20,21,22,23]. Two of them compared HW vs. non-HW periods [20,21]. Xu et al. performed subgroup analyses based on the HW definitions of the original studies—i.e., temperature above a predefined threshold, expressed as a percentile or cutoff, for two, three, or five days, respectively [20]. The relative risks (RRs) ranged from 1.03 (95% confidence interval [CI] 1.02–1.04; number of considered studies [N] = 3) to 1.20 (95% CI 1.02–1.41; N = 2) and were all statistically significant except for HWs defined as *“periods of five or more days with mean temperature higher than 97th percentile”*. Dimitrova et al. found an RR of 1.22 (95% CI 1.10–1.36; N = 5) when considering a 10 °C increase above 25 °C with a 0–1 day lag, and a RR of 1.23 (95% CI 1.11–1.37; N = 5) when considering a 5.5 °C increase above 26.5 °C with a 0–13 day lag [21]. The other two meta-analyses considered the effect associated with a 1 °C increase [22,23]. Luo et al. found an RR of 1.02 (95% CI 1.02–1.02; N = 44) [23], while Odame et al. reported an RR of 1.03 (95% CI 1.01–1.05; N = 10) [22].

#### 3.1.2. Mortality for Cardiovascular Diseases

In total, five meta-analyses dealt with mortality for cardiovascular diseases [20,22,24,25,26]. Three of them compared HW vs. non-HW periods [20,24,25]. Cheng et al. found an RR of 1.15 (95% CI 1.09–1.21; N = 36) during HWs [24]. In addition, the authors performed sub-group analyses finding an RR for ischemic heart disease of 1.23 (95% CI 1.07–1.42; N = 11) and an RR for heart failure of 1.10 (95% CI 1.04–1.18; N = 2). Xu et al. found an RR of 1.21 (95% CI 1.13–1.29; N = 2) when considering temperature above 35 °C for two or more days, and a RR of 1.09 (95% CI 1.06–1.12; N = 2) when considering temperature above 98th percentile for two or more days [20]. Sun et al. defined HWs as *“two or more days exceeding the defined temperature (e.g., 95th percentile)”* and found an RR for death caused by myocardial infarction of 1.64 (95% CI 1.09–2.47; N = 4) [25]. Two meta-analyses considered the effect associated with a 1 °C increase [22,26]. Moghadamnia et al. found an RR of 1.01 (95% CI 1.01–1.02; N = 33) [26], and Odame et al. found an RR of 1.11 (95% CI 1.05–1.18; N = 4) [22].

#### 3.1.3. Mortality for Respiratory Diseases

In total, three meta-analyses dealt with mortality for respiratory diseases [20,23,24]. Two meta-analyses compared HW vs. non-HW periods [20,24]. Xu et al. found an RR of 1.22 (95% CI 1.06–1.37; N = 2) when considering temperatures above 35 °C for three or more days, and a RR of 1.06 (95% CI 1.00–1.12; N = 2) when considering temperatures above 98th percentile for two or more days [20]. Cheng et al. generically considered HWs as *“temperatures higher than normal for a specified number of days”* and found an RR of 1.18 (95% CI 1.09–1.28; N = 25) [24]. In addition, the authors performed subgroup analyses finding an RR for chronic obstructive pulmonary disease of 1.14 (95% CI 1.08–1.21; N = 7). With regard to the meta-analysis that considered the effect associated with a 1 °C increase, Luo et al. found an RR of 1.02 (95% CI 1.01–1.03; N = 44) [23].

#### 3.1.4. Mortality for Renal Diseases

One meta-analysis dealt with mortality for renal diseases [27]. Liu et al. performed different meta-analyses according to the definitions of HW adopted by the original studies. When comparing HW vs. non-HW periods, the overall RR was 1.18 (95% CI 1.08–1.29; N = 36). Different subgroup analyses were performed classifying studies according to the percentiles for the specified thresholds (from the 90th to the 99th). Significant RRs were found only for the percentiles from 90th to 93rd (RR 1.24, 95% CI 1.01–1.48; N = 4), and from 90th to 99th (RR 1.19; 95% CI 1.08–1.30; N = 18). In addition, authors considered the effect associated with a 1 °C increase, finding an RR of 1.03 (95% CI 1.02–1.05; N = 42).

#### 3.1.5. Mortality for Cerebrovascular Diseases

Three meta-analyses dealt with mortality for cerebrovascular diseases [23,24,28]. Cheng et al. found an RR of 1.19 (95% CI 1.04–1.36; N = 14) for stroke mortality, generically considering HWs as *“periods with temperatures higher than normal for several days”* [24]. Two meta-analyses considered the effect associated with a 1 °C increase [23,28]. Luo et al. found an RR of 1.02 (95% CI 1.02–1.03; N = 44) for any cerebrovascular disease [23]; Lian et al. found an RR of 1.02 (95% CI 1.01–1.02; N = 20) for stroke mortality [28].

#### 3.1.6. Mortality for Mental Diseases

One meta-analysis dealt with mortality for mental illness [29]. Liu et al. considered the effect associated with a 1 °C increase and found an RR of 1.02 (95% CI 1.02–1.03; N = 12). In addition, two subgroup analyses were performed. The RR of death for mental and behavioral disorders was 1.03 (95% CI 1.01–1.05; N = 5), and the RR of death for suicide and self-harm was 1.01 (95% CI 1.00–1.02; N = 7).

#### 3.1.7. Mortality for Diabetes

Two meta-analyses dealt with mortality for diabetes [30,31]. Song et al. included studies considering different definitions of HWs and found an overall RR for diabetes-related mortality of 1.14 (95% CI 1.09–1.19; N = 13). Within the subgroup analyses, the RRs for diabetes-related mortality associated with temperatures above 90th and 99th percentile were 1.11 (95% CI 1.07–1.15; N = 3) and 1.24 (95% CI 1.16–1.33; N = 3), respectively. Additionally, the RR associated with a 1 °C increase was 1.04 (95% CI 1.02–1.07; N = 5) [30]. Moon found an RR of death in diabetic patients of 1.18 (95% CI 1.13–1.25; N = 25) considering HW vs. non-HW periods [31].

#### 3.1.8. Mortality for Injuries

While no meta-analysis evaluated mortality for injuries, two systematic reviews addressed this topic [32,33]. Levi et al. included 165 articles and reported that *“extreme heat”* was generally associated with an increased risk of death from injury among workers especially in the agricultural and construction sectors [32]. Bonafede et al. reported among others an odds ratio (OR) of 2.32 (1.55–3.48) for those working in the construction sector, an OR of 3.50 (1.94–6.32) for those working in the agricultural sector and an OR of 10.17 (5.38–19.43) for those of unknown sector [33].

### 3.2. Morbidity

#### 3.2.1. Morbidity for Cardiovascular Diseases

Four meta-analyses dealt with cardiovascular morbidity and used hospitalization as the outcome [24,25,34,35]. Cheng et al. also included emergency department (ED) visits and ambulance attendances/call-outs/transports as outcomes [24]. Phung et al. considered HWs as *“periods equal to or longer than two days with extreme temperatures (e.g., above the 95th percentile)”* and found an RR of 1.02 (95% CI 1.01–1.04; N = 23) [34]. Cheng et al. found an RR of 1.00 (95% CI 1.00–1.00; N = 18) generically considering HWs as *“periods with temperature higher than normal and lasting several days”* [24]. For every 1 °C increase, Turner et al. found an RR for cardiovascular hospitalization of 1.00 (95% CI 0.98–1.02; N = 16) [35], while Sun et al. found an RR of 1.02 (95% CI 1.00–1.03; N = 13) for hospitalization due to myocardial infarction [25].

#### 3.2.2. Morbidity from Respiratory Diseases

Two meta-analyses dealt with respiratory morbidity [24,35]. Both studies used hospitalization as the outcome, but Cheng et al. considered also emergency department visits and ambulance attendances/call-outs/transports [24]. Cheng et al. found an RR of 1.04 (95% CI 1.00–1.09; N = 17) generically considering HW as periods with temperature higher than normal and lasting several days. In addition, two subgroup analyses were performed, finding an RR of 1.04 (95% CI 0.93–1.17; N = 2) for chronic obstructive pulmonary disease and an RR of 1.00 (95% CI 1.00–1.00; N = 3) for asthma [24]. Turner et al. found an RR of 1.02 (95% CI 0.99–1.06; N = 110) for overall respiratory hospitalization for every 1 °C increase [35].

#### 3.2.3. Morbidity for Renal Diseases

Four meta-analyses dealt with renal morbidity [27,36,37,38]. All studies used hospitalization and emergency department visits as outcomes. Liu et al. also used ambulance call-outs [27], while Zhang et al. considered records for generic medical consultations and extracorporeal shock wave lithotripsy [38]. Lee et al. and Liu et al. compared HW vs. non-HW periods using temperature thresholds and 90th-99th percentiles, finding an RR of 1.24 (95% CI 1.19–1.28; N = 16) and an RR of 1.06 (95% CI 1.04–1.07; N = 37), respectively [27,36]. In addition, Liu et al. assessed renal morbidity for each 1 °C increase, finding an RR of 1.01 (95% CI 1.01–1.01; N = 35) [27]. Three meta-analyses assessed the association between HWs and urolithiasis. Lee et al. (2019) found an RR of 1.32 (95% CI 1.24–1.40; N = 16) during HWs [36], while Zhang et al. found an RR of 1.05 (95% CI 1.04–1.06; N = 12) for every 1 °C increase [38]. Finally, Liu et al. found an RR of 1.02 (95% CI 1.02–1.03; N = 22) for every 1 °C increase [27]. Two meta-analyses assessed the association between HWs and acute kidney injury. Flouris et al. found an RR of 1.15 (95% CI 1.11–1.19; N = 10) while working in heat stress conditions [37]. Liu et al. found an RR of 1.01 (95% CI 1.01–1.02; N = 25) for every 1 °C increase [27]. In addition, Liu et al. assessed the risk of urinary tract infections and kidney failure for each 1 °C increase, finding an RR of 1.01 (95% CI 1.00–1.01; N = 3) and of 1.01 (95% CI 1.01–1.01) for the two conditions [27].

#### 3.2.4. Morbidity for Cerebrovascular Disease

Three meta-analyses dealt with cerebrovascular morbidity [24,28,35]. Both studies used hospitalization and ED visits as outcomes, but Cheng et al. also used ambulance call-outs [24]. Cheng et al. compared HW vs. non-HW periods and found an RR of 0.99 (95% CI 0.96–1.02; N = 7) [24], while Turner et al. focused on stroke morbidity, finding an RR 0.99 (95% CI 0.89–1.11; N = 5) [35]. Lian et al. assessed the risk of the occurrence of a major adverse cerebrovascular event for each 1 °C increase, finding an RR of 1.01 (95% CI 1.01–1.02; N = 27) [28]. In addition, subgroup analyses found an RR of 0.98 (95% CI 0.96–1.00; N = 4) for hemorrhagic stroke and an RR of 1.01 (1.00–1.02; N = 10) for ischemic stroke.

#### 3.2.5. Morbidity for Mental Diseases

One meta-analysis dealt with morbidity from mental illness and used hospitalizations and ED visits as outcomes [29]. Overall, the RR for mental illness morbidity for every 1 °C increase was 1.01 (95% CI 1.01–1.01; N = 18). A subgroup analysis for mental and behavioral disorders morbidity found an RR of 1.01 (95% CI 1.01–1.01; N = 12).

#### 3.2.6. Morbidity for Injuries

Three meta-analyses dealt with occupational injuries morbidity [37,39,40]. Only Fatima et al. clearly defined occupational injuries and referred to them as *“any personal injury (injury, illness, or death) taking place at the workplace as the result of an occupational accident”* [40]. Binazzi et al. included different measurement approaches for heat exposure (specifically: daily maximum and minimum or average temperature; maximum daily humidity index; maximum wet bulb temperature; 1 °C increase in maximum temperature; or ≥60th–95th percentile in daily maximum and minimum temperatures) and found an RR of 1.01 (95% CI 1.00–1.01; N = 6) considering *“exposure to the highest temperature”* [39], while Flouris et al. found an RR of 1.35 (95% CI 1.31–1.39; N = 33) [37]. Fatima et al. analyzed HW vs. non-HW periods, finding an RR of 1.17 (95% CI 1.06–1.29; N = 8), and the association with a 1 °C increase, finding an RR of 1.01 (95% CI 1.01–1.01; N = 17) [40].

#### 3.2.7. Morbidity for Childbirth Diseases

One meta-analysis dealt with childbirth morbidity [41]. The study assessed the impact of HWs on stillbirth and preterm birth, defined as a live birth before the 37th week of gestation. For preterm birth, the authors found an OR of 1.16 (95% CI 1.10–1.23; N = 6) with HWs generically defined as *“two or more days with temperatures above a predefined threshold”*; they found an OR of 1.01 (95% CI 1.01–1.02; N = 21) when considering a period of four or more weeks with high temperatures. When considering the effect associated with a 1 °C increase, an OR of 1.05 (95% CI 1.03–1.07; N = 7) was found. Chersich et al. [41] performed a meta-analysis on stillbirths, finding an OR of 1.24 (95% CI 1.12–1.36; N = 4) when considering periods equal to or shorter than 1 week with high temperatures, and an OR of 1.05 (95% CI 1.01–1.08; N = 3) when considering the effect associated with a 1 °C increase.

#### 3.2.8. Morbidity for Diabetes

Two meta-analyses dealt with diabetes morbidity [30,31]. Moon found an RR of 1.10 (95% CI 1.06–1.14; N = 15) for hospital admissions, ED visits, general practitioner consultations, or morbid symptoms, in individuals with diabetes during HWs [31]. Song et al. found an RR of 1.11 (95% CI 1.03–1.19; N = 3) for diabetes morbidity in periods with temperatures higher than the 99th percentile and an RR of 1.01 (95% CI 1.01–1.01; N = 4) when considering the effect associated with a 1 °C increase [30].

#### 3.2.9. Morbidity for Infectious Diseases

One meta-analysis evaluated Hand-Foot-Mouth disease and found an RR of 1.11 (95% CI 1.08–1.13; N = 23) for each 1 °C increase [42].

### 3.3. Vulnerability

#### 3.3.1. Pre-Existing Health Conditions

One meta-analysis evaluated the role of pre-existing health conditions on the increase in general mortality during HWs. The considered conditions were cardiovascular diseases (OR 2.48; 95% CI 1.3–4.8; N = 4), pulmonary diseases (OR 1.61; 95% CI 1.2–2.1; N = 3), mental illness (OR 3.61; 95% CI 1.3–9.8; N = 5), and dependence on psychotropic medications (OR 1.9; 95% CI 1.3–2.8; N = 4) [43].

#### 3.3.2. Demographic Factors

Three meta-analyses evaluated the role of age on the increase in risk of detrimental health effects during HWs [23,28,44]. Benmarhnia et al. compared HW vs. non-HW periods, finding an interaction RR (iRR) of 1.02 (95% CI 1.01–1.03; N = 39) for overall mortality in subjects aged at least 65 years, compared to younger ones [44]. Two meta-analyses assessed the effect modification by age associated with a 1 °C increase. Luo et al. found an RR of 1.03 (95% CI 1.02–1.04; N = 44) for overall mortality among the elderly (aged more than 65 years), while they found an RR of 1.01 (95% CI 1.01–1.02; N = 44) among younger individuals [23]. Lian et al. found an RR of 1.01 (95% CI 1.00–1.02; N = 7) for stroke among the elderly (aged at least 65 years), while they found an RR of 1.00 (95% CI 1.000–1.00; N = 7) among younger individuals [28]. Three meta-analyses evaluated the role of gender in the increase of general mortality during HWs. Benmarhnia et al. included in their meta-analysis different definitions of HW, finding a non-significant iRR of 0.99 (95% CI 0.97–1.01; N = 39) for overall mortality among men, compared to women [44]. Luo et al. found an RR of 1.02 (95% CI 1.02–1.03; N = 44) for overall mortality among men, compared to an RR of 1.03 (95% CI 1.02–1.04; N = 44) among women [23]. Lian et al. found an RR of 1.02 (95% CI 1.00–1.03; N = 7) for stroke among men, while the correspondent RR among women was 1.02 (95% CI 0.99–1.05; N = 7) [28].

#### 3.3.3. Environmental Factors

Three meta-analyses evaluated the role of environmental factors in the increase of mortality/morbidity during HWs [23,43,45]. Bouchama et al. compared HW vs. non-HW periods, finding an OR of 4.35 (95% CI 1.67–10.0; N = 6) for subjects without air conditioning at home and an OR of 1.67 (95% CI 0.91–2.5; N = 3) for not having a fan [43]. Schinasi et al. evaluated urban microclimate and found that residents in hotter areas within cities had an RR of 1.06 (95% CI 1.03–1.09; N = 6) of mortality/morbidity compared to those in cooler areas. Moreover, living in less vegetated areas was associated with an RR of 1.05 (95% CI: 1.00–1.11; N = 6) compared to living in more vegetated areas [45]. Luo et al. instead assessed the effect on overall mortality associated with a 1 °C increase in different climates: they found an RR of 1.01 (95% CI 1.01–1.02; N = 44) for dry climate, an RR of 1.03 (95% CI 1.03–1.04; N = 44) for temperate climate, and an RR of 1.01 (95% CI 1.01–1.02; N = 44) for the continental climate [23].

#### 3.3.4. Social and Economic Factors

Three meta-analyses evaluated the role of social factors and socioeconomic status (SES) on the increase of general mortality during HWs [23,43,44]. In particular, the following social factors were considered: living alone, being confined to bed, being unable to care for oneself, leaving home, and having social contacts. Benmarhnia et al. compared HW vs. non-HW periods, finding an interaction RR (iRR) of 1.03 (95% CI 1.01–1.05; N = 15) for overall mortality in subjects with low SES (measured at the individual level), compared to those with high SES. When SES was considered at the aggregate (ecological) level, the corresponding iRR was 1.01 (95% CI 1.99–1.02; N = 12) [44]. Bouchama et al. found that living alone (OR = 2.09; 95% CI 0.7–6.5; N = 4), not leaving home (OR = 3.35; 95% CI 41.6–6.69; N = 4), being confined to bed (OR = 6.44; 95% CI 4.5–9.2; N = 3), and not being able to care for oneself (OR = 2.97; 95% CI 1.8–4.8; N = 4) were associated with an increased risk of death during HWs, while a protective effect was found for increased social contacts (OR = 0.4; 95% CI 0.2–0.8; N = 4) [43]. Luo et al. instead assessed the effect on overall mortality associated with a 1 °C increase in China, finding an RR of 1.04 (95% CI 1.01–1.08; N = 44) among individuals with low SES, compared to an RR of 1.04 (95% CI (1.01–1.07); N = 44) among those with high SES [23].

### 3.4. Adaptation

#### 3.4.1. Policies, Plans, and Interventions

Ten systematic reviews analyzed policies aimed at reducing the impact of HWs on communities, summarizing recommendations to advance future research, and pointing out gaps and limitations in current mitigation strategies [46,47,48,49,50,51,52,53,54,55]. Overall, most of the considered articles used mortality and morbidity as outcomes, with limited investigations on economy-related aspects. Schmitt et al. underline the lack of research on the economic impact—especially in the long-term—and on the effectiveness of interventions [47], while Palinkas et al. indicate the lack of specific strategies to deal with mental health issues associated with HWs [53]. The variety of possible public health interventions that are discussed clearly lacks a formal classification: several authors try to summarize the gathered evidence according to a specific aim, such as structural interventions on buildings and city public spaces [46,49,50,52]; others instead focus on the target group, for example interventions dedicated to vulnerable strata of the population [46,51]. Mayrhuber et al. divide interventions into those aiming to detect or influence risk and those targeting protective factors [51], while Boeckmann et al. differentiate heat prevention actions that are structural vs. individual [49].

#### 3.4.2. Heat Warning Systems

Five systematic reviews addressed a specific type of public health intervention, namely the Heat Health Action Plans and the pertaining Heat Warning Systems [46,48,49,55,56]. Martiello et al. described the importance of Heat Warning Systems in potentially preventing outdoor working injuries [55]. Lundgren Kownacki et al. instead pointed out that most Heat Warning Systems are based on predicted outside temperature while they should also consider indoor temperature, as this is where most people usually stay during HWs [56]. In their systematic review of reviews on this topic, Bouzid et al. found that studies were often based on low quality data or had important limitations [46]. For example, some studies did not include the most vulnerable strata of the population (e.g., the elderly or fragile individuals), thus limiting the external validity of their conclusions. In the same vein, others attributed a reduction in overall mortality to HWs before and after the implementation of Heat Warning Systems without accounting for possible confounders. For example, Boeckmann et al. found 17 observational studies comparing the effect of Heat Warning Systems with a before/after approach, which does not consider changes in the prevalence of confounders over time (e.g., increase in use of air conditioning). This makes it difficult to identify clear associations between interventions and effective reduction in negative health outcomes [49].

### 3.5. Risk Perception

As anticipated, evidence on risk perception of HWs was particularly scanty, and no meta-analysis was available. Six systematic reviews assessed risk perception related to HWs [46,49,51,57,58,59]. All the reviews agreed on the fact that risk perception varies with demographic, social, and economic characteristics. Elderly were considered as a group with a particularly low-risk perception, despite being a high-risk population [46,49,51,57,59]. In contrast, young people and populations at a higher educational level were more aware of the heat-related risks [59]. Other factors affecting the risk perception were income, education, and trust in the source of information used [58].

## 4. Discussion

The public’s concern regarding the effects of climate change and, specifically, of HWs is growing, and in the last years different governments have enacted measures aiming to reduce the burden of greenhouse gas emissions and have signed international agreements. Unfortunately, we are still far from achieving carbon neutrality, and a clear roadmap detailing the actions of each country is currently missing [60]. Against this background, our planet continues to warm up, and HWs are globally increasing and intensifying [16]. Considering the above, it is important to improve the quality of the scientific literature on the effects of HWs on health, to ultimately contribute to implementing effective, evidence-based interventions.

As far as the authors know, this is the first systematic review of published reviews mapping the available evidence on the health effects of HWs. This complements the results of a recently published overview of systematic reviews investigating the health effects of climate change in general [61]. It is noteworthy that although the latter had a broader scope than the present review, the authors found a smaller number of studies. This suggests that some papers dealing with specific effects of climate change, such as HWs, might be overlooked when focusing on climate change in general.

Overall, our results highlight the large number of published reviews on HWs and health in recent years. Possibly, such a growing interest in this topic has also been fostered by the increase in intensity and frequency of HWs that occurred in the last decade [13] and by the mounting awareness on climate change [62]. Nevertheless, this review of reviews underlines some important limitations of the HW-related body of evidence, which risk hampering research and policy advancements.

The first important limitation is the heterogeneity of HW definitions as well as the different thresholds that are used to define these phenomena. Without a universally shared definition it is difficult to compare studies and to develop evidence-based policies. Although at a local level HWs and response plans adopt operational definitions of an HW based on specific contexts [63,64,65], the use of a common definition would play a pivotal role for understanding HW risk at a global scale and for the subsequent risk reduction plans [66].

Another important limitation concerns the scarce attention paid by literature syntheses to the impact of HWs on mental health outcomes. Although climate change has been recognized as a factor affecting mental health [67], a systematic and comprehensive assessment of the psychosocial impact of HWs is currently missing, corroborating the findings from Rocque et al. [61]. The consideration of mental health aspects in HWs research is crucial, as HWs might trigger the onset of new mental disorders, as well as exacerbate pre-existing conditions. Many social determinants of health may also play a role in this and mediate the impact of HWs on mental health. Comprehensive syntheses on these aspects may shed light on interesting findings. Future reviews summarizing evidence on psychotropic drug consumption during HWs, as well as syntheses of studies on mental health services utilization during HWs could be useful to advance our understanding of these issues.

Only a few reviews and no meta-analyses were found on the risk perception of HWs. The link between risk perception and population response to heat warnings and the consequent implementation of adaptive measures has been demonstrated by previous studies, with social and economic factors influencing perception and adaptation [59]. The conduction of quantitative and qualitative research syntheses on this aspect is warranted, to provide not only the basis for understanding the population’s coping behavior, but also to set the basis for the establishment of effective warning systems and interventions to improve community preparedness.

Besides the focus on Heat Warning Systems, the body of evidence on other interventions or public health measures employed to mitigate the impact of heatwaves on health is scarce. Given the recent interest from governments and international agencies in climate change adaptation [15,16], it is essential that the scientific community is mobilized to provide the necessary knowledge base to pave the way for future interventions [68].

Key messages regarding knowledge gaps and recommendations have been summarized in Figure 5. This visual summary aims to provide a broad but comprehensive overview of the knowledge gaps and the research priorities on the health effects of HWs in a form that can be accessible also by the non-academic reader, with the goal of contributing to reduce the hiatus between scientific evidence and policy making [69,70].

Some limitations of our work should be noted. First, reviews written in languages different from those listed in the inclusion criteria might have added further information. However, the proportion of articles excluded for language purposes is very small (less than 1% of records assessed for eligibility) and therefore we are reasonably confident that we did not miss any important information. Second, we did not summarize the information reported by non-systematic reviews. This was decided to give priority to higher quality studies, namely meta-analyses and systematic reviews, bearing in mind the urgency to provide policymakers with a practical translation of evidence-based information. Third, while this review is focused on HWs, it should be taken into account that cold spells are very relevant as well. In fact, deaths due to cold spells exceed those due to HWs in many geographical areas [71]. A future review of reviews on the effect of cold temperature is thus warranted to complement the results of the present research. Finally, as the majority of the reviews encompassed original studies from different regions and settings, we did not assess their geographical distribution. However, it is reasonable to recommend an improvement in research initiatives targeting areas at particularly high risk of HWs in the future, such as low- and middle-income countries, and South-Eastern Europe [14].

## 5. Conclusions

Climate change is a prominent global health issue that humankind is currently facing. Among its multifaceted manifestations, HWs are significantly influencing several aspects of global health. Nevertheless, the body of systematic syntheses on their impacts on health is still inconsistent and patchy. We believe that future research should focus on filling the gaps summarized in this first ever systematic review of published reviews on this topic to provide sound evidence and support global public health interventions. Specifically: establish a universal operational definition of HWs; advance research on morbidity and health systems’ strain consequent to HWs; quantitatively investigate vulnerability to HWs; improve the quality of research on Heat Health Action Plans and Heat Warning Systems; and assess HW-related risk perception with mixed-methods syntheses. By delivering a simple tool that is also easily accessible to stakeholders without a healthcare or public health background, we envision overcoming the existing barriers between science and policy making.

## Figures and Tables

**Figure 1 ijerph-19-05887-f001:**
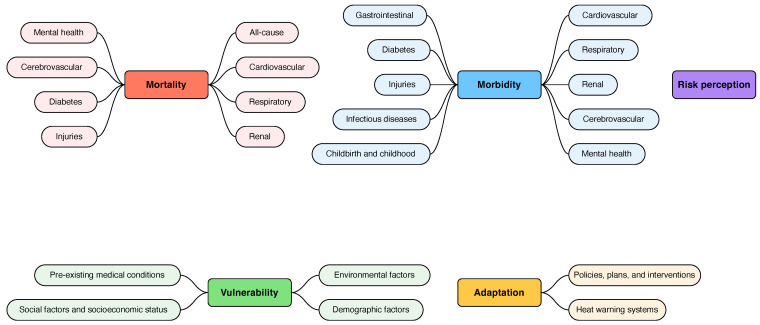
Categories and topics.

**Figure 2 ijerph-19-05887-f002:**
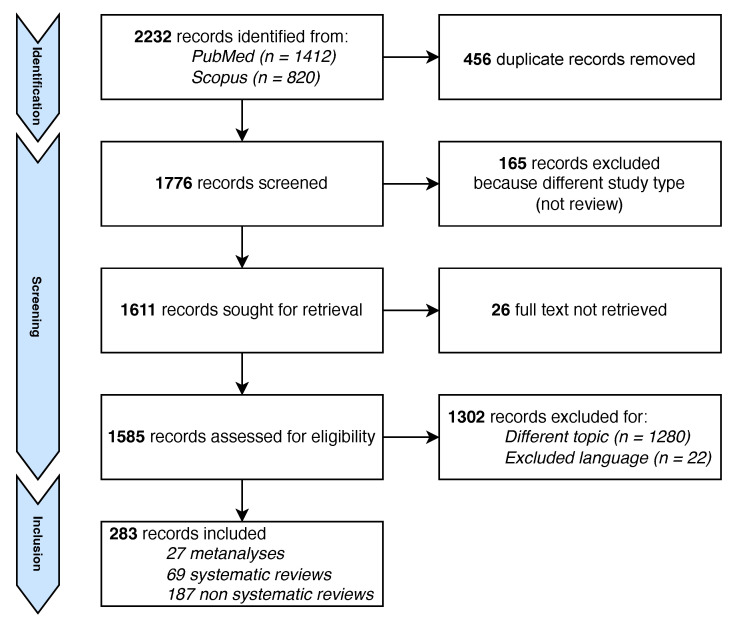
PRISMA flowchart.

**Figure 3 ijerph-19-05887-f003:**
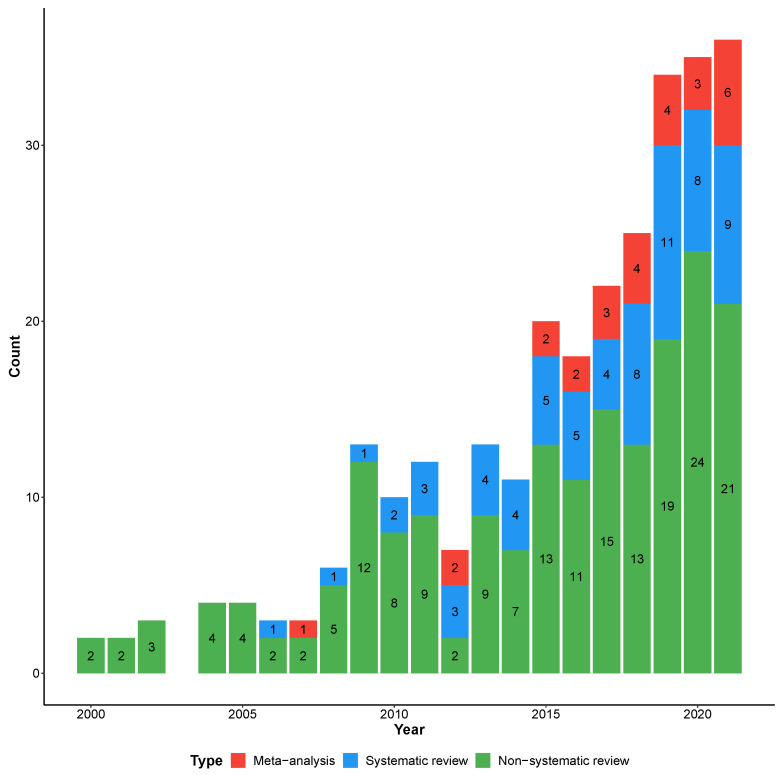
Type of review over the years.

**Figure 4 ijerph-19-05887-f004:**
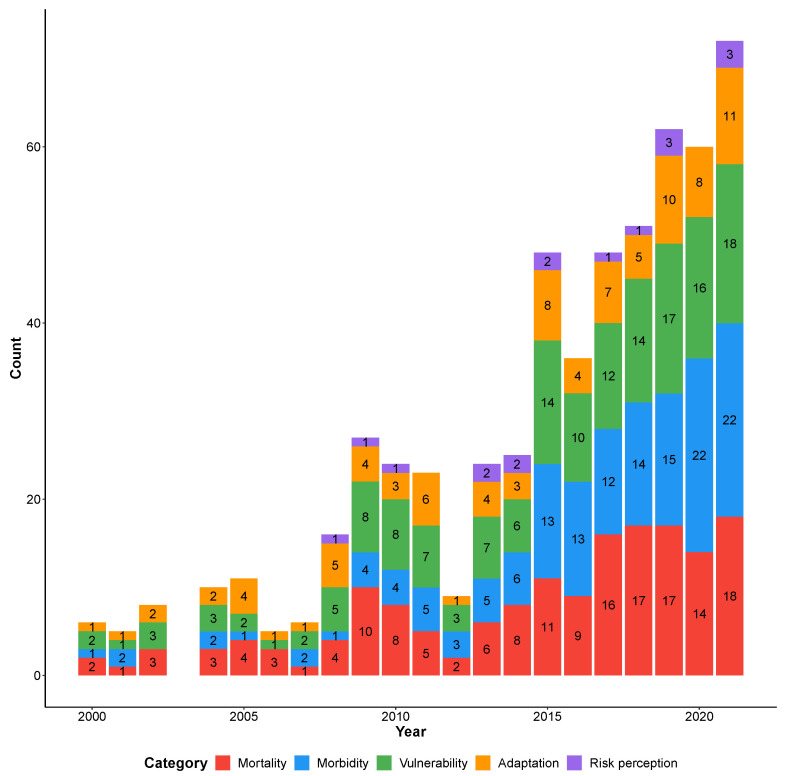
Categories addressed over the years.

**Figure 5 ijerph-19-05887-f005:**
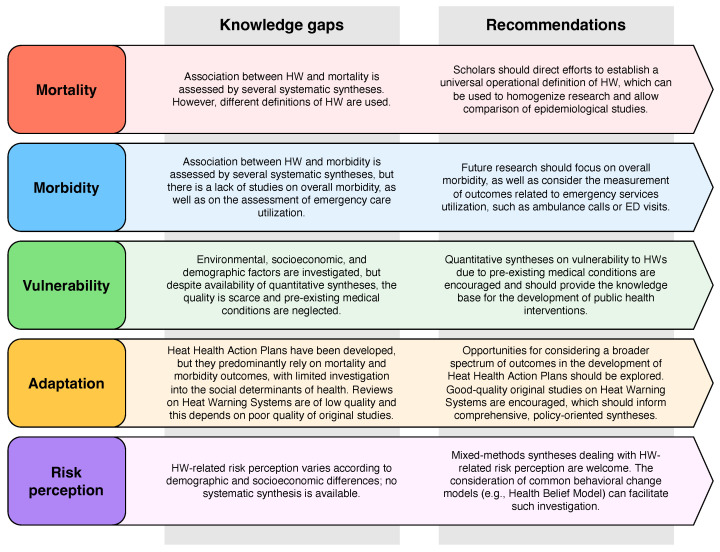
Knowledge gaps and recommendations emerged from the included reviews.

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
