# Peer review of "Knowledge Gaps and Research Priorities on the Health Effects of Heatwaves: A Systematic Review of Reviews"

_ijerph, 2022, doi:10.3390/ijerph19105887_

Round 1

Reviewer 1 Report

Authors have done a good job in reviewing available knowledge on the effects of heat waves. Gaps have been clearly identified, hopefully triggering useful new research. The manuscript certainly deserves to be published. However I noticed a few topics that would make the context and framing more complete. Please see my comments below, sorted by line number. There have been phases of frequent heat waves also before, in pre-industrial times. So any findings for the modern times may also apply to some Medieval warm times in Europe and other heat anomalies. I appreciate that the authors workd on heat waves. However, the threat of cold to humans is left out. It should be mentioned at least in the introduction for completeness. I noticed a few other style issues which I discuss below.

Lines 1 and 17: „As a consequence of modern global warming…“ There have been also pre-industrial warming phases, e.g. Luterbacher et al. 2016: https://iopscience.iop.org/article/10.1088/1748-9326/11/2/024001

Maybe worth mentioning somewhere in the introduction. Another paper is by Büntgen et al. 2020: https://www.sciencedirect.com/science/article/pii/S1125786520300965 See their Fig. 5 for colour stripes ove the past 2000 years. People have repeatedly suffered from phases with more frequent heat waves during the past 2000 years. In the pre-industrial past they did not even have the chance to protect themselves with air condition.

Lines 21-25: World population is also growing fast, mostly in the hotter regions of the world. This needs to be mentioned here as it contributes to the rise of people vulnerable to heat waves.

Lines 34-40: Would it make sense to compare average life expectancy of inhabitants of hot and cold countries? https://www.worlddata.info/life-expectancy.php

Qatar has 79.1 years for males and 82 years for females.

Czechia has 76.3 years for males and 81.1 for females.

Just two random examples. How do people in Qatar manage to adapt to heat so well? I guess it is general availability of air condition. Could this be a strategy to equip all households with a/c to be prepared for summer heat waves?

Introduction: I think you should also mention briefly deaths by cold waves, and the significance of heat vs cold deaths. Curretnly cold deaths are still dominating over heat deaths. See: Ingole et al. 2022: https://www.sciencedirect.com/science/article/pii/S0013935121016054

Press release: https://phys.org/news/2022-01-cold-deaths-india.html

Line 383: authors write: „extreme weather events are increasing and intensifying“. This statement is not valid for all extreme weatehr types, therefore it is wrong to generalize this. Check the IPCC AR6 report more thoroughly: Storms, droughts, floods are NOT increasing in many parts of the world. The only extreme weather event that actually is increasing globally are heat waves. As this paper is about heat waves, it might be safe to limit the statement to this extreme weather type.

Line 408: Increasing violence during hot days is partly related to the fact that people spend more time outdoors. It is really tricky to differentiate the various factors here. It would be simplistic to say that people’s mental health deteriorates with increasing heat. Are inhabitants of hotter countries more aggressive and affected than people from cold countries? It is somewhat risky to make such statements, as it may be misunderstood as some sort of racism or discrimination. Some newspapers have tried to spread such messages („heat waves make people mad“). It would be important to avoid such generalizations and stick to the neutral science.

Line 442: „Climate change is one of the worst threats that humankind is currently facing.“ Whilst this fits into the standard narrative, it should be avoided to make such statements in scientific publications. The Russian invasion into the Ukraine and threat of a new nuclear war has really reminded us that there are other threats that are at least as threatening as climate change. The time scale is even much shorter, the danger more imminent. At present, high inflation is causing huge problems for many people. Availability of key goods has become a problem. The corona pandemic has reminded us of the threat of virus and other deseases. Alarmist slang should be avoided in scientific papers.  

Author Response

Dear Reviewer, we would like to thank yo for the effort in revising the manuscript and providing suggestions for its improvements. Please see the attachment for our responses.

Reviewer 2 Report

Dear authors,

The study is intriguing, the method chosen for the study is appropriate, and the results are valuable. Even so, several studies with high visibility are missing from the analysis and are valuable with a high impact on the study area. Further, I recommend several:

  1. Mora, C., Spirandelli, D., Franklin, E. C., Lynham, J., Kantar, M. B., Miles, W., ... & Hunter, C. L. (2018). Broad threat to humanity from cumulative climate hazards intensified by greenhouse gas emissions. Nature Climate Change, 8(12), 1062-1071.
  2. Li, X. X. (2020). Heat wave trends in Southeast Asia during 1979–2018: The impact of humidity. Science of The Total Environment, 721, 137664.
  3. Stillman, J. H. (2019). Heat waves, the new normal: summertime temperature extremes will impact animals, ecosystems, and human communities. Physiology, 34(2), 86-100.
  4. Yang, J., Yin, P., Sun, J., Wang, B., Zhou, M., Li, M., ... & Liu, Q. (2019). Heatwave and mortality in 31 major Chinese cities: definition, vulnerability and implications. Science of The Total Environment, 649, 695-702.
  5. Guo, Y., Gasparrini, A., Li, S., Sera, F., Vicedo-Cabrera, A. M., de Sousa Zanotti Stagliorio Coelho, M., ... & Tong, S. (2018). Quantifying excess deaths related to heatwaves under climate change scenarios: A multicountry time series modelling study. PLoS medicine, 15(7), e1002629.

In the introduction, please add information regarding the last year's occurred heat waves in the world. For example, I recommend a synthesis of reported studies, especially after 2018, which were called the Lucifer heatwave (Guerreiro, S. B., Dawson, R. J., Kilsby, C., Lewis, E., & Ford, A. (2018). Future heat-waves, droughts and floods in 571 European cities. Environmental Research Letters, 13(3), 034009; Ma, F., Yuan, X., Jiao, Y., & Ji, P. (2020). Unprecedented Europe heat in June–July 2019: Risk in the historical and future context. Geophysical Research Letters, 47(11), e2020GL087809.).

Further, please indicate clearly the research area. It appears only in Methods as criteria of search articles. but a statement must be included even in the introduction chapter.

L38. Please delete "With this aim in mind"

At the end of the Introduction chapter please add a statement regarding the hypothesis and the main aim of the study must be clearly presented. In the present form, it appears to be very light reported on the information existing in the results and especially not connected with the title of the article.

The Conclusion needs more strength. A lot of information is presented in the results and some are very interesting. Please include the most important in a conclusion from which to be obvious that the heatwaves are influencing not only the quality of life but even threaten the life itself. Considering the fact that the analyzed countries are strongly developed economically with well-defined insurance systems, I think that a recommendation can be formulated from this point of view as well. Moreover, based on this analysis, conclusions can be drawn regarding the developing countries in Europe, especially in South-Eastern Europe, the area in which the heat waves are expected to be severe in the coming years.

Author Response

(The authors gave the same response as above.)

Round 2

Reviewer 2 Report

Dear Authors,

Thank you for considering the recommendations. The manuscript now has a more structured message.